# Research on Township Industry Development under GEP Accounting—A Case Study of Hanwang Town in Xuzhou City

**Shuai Tong [1], Jianjie Gao [2], Fengyu Wang [3] and Xiang Ji [4],***

1   School of Mechanics and Civil Engineering, China University of Mining and Technology, Xuzhou 221116, China; tb22030012a41@cumt.edu.cn
2   Northwest Branch, China Academy of Urban Planning and Design, Lanzhou 730000, China; gjj678@126.com
3   School of Architecture and Design, China University of Mining and Technology, Xuzhou 221116, China; ts21190007a31@cumt.edu.cn
4   Jiangsu Collaborative Innovation Center for Building Energy Saving and Construction Technology, School of Mechanics and Civil Engineering, China University of Mining and Technology, Xuzhou 221116, China
*   Correspondence: jixiang0615@yeah.net; Tel.: +86-139-0521-7566

**Abstract:** The protection and utilization of ecological environment are very important for urban and rural development. At present, a large number of relevant theoretical and practical explorations have been carried out, which confirms the important conclusion that lucid waters and lush mountains are invaluable assets. The sustainable development of ecological environments is based on coordination with human production and life. In this paper, by constructing an accounting system for the gross ecosystem product (GEP) applicable to Hanwang town, using the market value method, the alternative cost method, the travel cost method, the willingness to pay method and other technical methods, the GEP of Hanwang town is calculated from three aspects: product supply, regulation service and cultural tourism. Finally, the spatial distribution characteristics of value are used to guide the development and layout of ecological industry in Hanwang town. The results showed that the total ecosystem product value of Hanwang town in Xuzhou was relatively high, reaching 1.165 billion CNY, with per capita reaching 30 million CNY, which was 49.16% of the town's GDP in 2020. The value of cultural tourism is 820 million CNY, the value of regulatory services is 239 million CNY, and the value of product provision is 106 million CNY. The ecological value of Hanwang town varies greatly in spatial distribution. On the whole, the price is low in the southwest, but high in the northeast. The high-value areas are mainly concentrated in three areas: Yudai River Riverside, Xuzhou Paradise in the north, Hanwang Scenic Spot in the middle and the Panaxi Valley tourist spot in the south. Based on the principle of ecological value transformation, combining with the spatial distribution characteristics of ecological value in Hanwang town, four modes of promoting ecological value transformation were proposed: ecological industrialization management, ecological governance and value promotion, ecological resource index trading and ecotourism. This paper preliminarily explores a method to calculate and transform the value of ecological space, which provides feasible concrete strategies for the protection of ecological space and the development of ecological industry in towns.

**Keywords:** gross ecosystem product (GEP); transformation of ecological value; ecological industry planning strategy; Hanwang town

## 1. Introduction

On the basis of industrial civilization, Chinese urban and rural development has made great achievements. However, rapid urbanization also brings ecological environment problems. Since the 18th CPC National Congress, China has attached great importance to the construction of ecological civilization [1]. Therefore, on the premise of fully recognizing the guiding role of ecological environments on planning, the control and constraint function of territorial spatial planning on space development and protection can be better exerted [2].

Sustainable development requires the coordinated development of urbanization and ecological environments [3]. The development of urbanization also needs to take the road of intensive, smart, green and low carbon [4]. According to the general requirements of the rural revitalization strategy, we need to build the ecological pattern of the countryside, and protect the natural resources, mountain features, landforms and biodiversity of the countryside. It is also needed to rationally divide the three-life space and build a coordinated life community of mountains, rivers, forests, fields, lakes and grasses [5]. Industrial development is the key to the strategy of rural revitalization, so, it is necessary to calculate the GEP as an important basis to guide the development of township ecological industry.

GEP accounting can be traced back to GDP accounting. Western scholars believe that economic accounting only considering GDP development is incomplete, because energy consumption and environmental damage will have a negative impact on economic value [6,7]. Huber [8] firstly proposed the concept of ecological compensation payment in 1978. There are also many practices related to ecological compensation in various countries, such as the United States Wetland Mitigation Bank [9], the PSA environmental payment program in Costa Rica [10], the Ecuador Basin Ecological Compensation Fund [11], Yangtze River Basin ecological compensation [12], China's conversion of farmland to forest, and Mexico's hydrological and ecological project [13,14]. Since the 1990s, ecologists began to study the function of ecosystem services [15]. In 1997, a study on the value of the world's ecosystems and natural assets was published to provide a reference standard and basis for related research [16]. On the basis of Costanza's research [17], a Chinese scholar made innovations and refinements [18] and proposed the equivalent factor table of service value per unit area of China's terrestrial ecosystems, which is applicable to China's eco-environmental value assessment [19]. This calculation method [20,21] has been widely used in many subsequent domestic studies [22,23].

In 2013, Chinese scholars firstly proposed the concept of "Gross Ecosystem Product" [24], which refers to the sum of the value of final products and services provided by the ecosystem for human welfare and sustainable economic and social development. The birth of the concept of GEP indicates that the research of ecological environment value in China has raised to a new level. Additionally, in 2021, it has been included in the United Nations' latest system of environmental and economic accounting, the Framework for Ecosystem Accounting (SEEA-EA). Some studies use emergy theory and methods to calculate the value of ecological environments in four parts [25]. Later GEP accounting researches incorporated eco-support services into the regulation services content, so it went from four parts to three parts, including value of product provision, regulation services and cultural tourism [26]. The GEP of the Qinghai Province [27], Ganzi Tibetan [28], Dalian City [29] and Xishui County [30] were evaluated and compared follow this method. In terms of ecological value transformation cases, there are many mature cases abroad including natural capital cost–benefit analysis in the United Kingdom, ecological benefit and ecological credits in Germany [31] and wetland mitigation bank in the United States.

Ecological value transformation cases in China are mainly concentrated in Fujian and Zhejiang Province because of their rich ecological resources and advanced economy. In domestic researches, ecological restoration, ecological product management, ecological index trading and ecological compensation constitute the most important four methods for ecological value transformation. The ecological value transformation strategy has been widely discussed in China, because it has been believed that transformation can bring win-win benefits about economic and environmental, whose logic is the most important [32]. However, there will be various limiting factors in the actual project, so it is necessary to consider the choice of strategy according to the situation. In particular, some ecological protection restrictions make it difficult to play the role of providing services and cultural services [33]. In the transformation process, we should pay special attention to the process method [34], but also pay attention to the innovation of special ecological products [35]. There are also some studies that introduced ecosystem services research into the field of urban ecosystem health research [36,37], broadening the research scope. Later, the

research area began to expand to fine space, shifted from macro region to the micro village area [38]. From the above analysis, it can be found that few studies use the results of GEP accounting to guide the development of township industries. In addition, there are few researches making GEP accounting from the perspective of administrative town, so as to guide ecological protection and exploitation. Therefore, it is necessary to explore the path, which is an important eco-oriented development innovation in ecological industry.

Firstly, this paper establishes the GEP accounting standard applicable to the area, and then calculates the ecological environment value by taking Hanwang town in Xuzhou as the case study. After that, the distribution characteristics of ecological value of Hanwang town are analyzed. Finally, the development strategies of various ecological industries within the town are determined according to the framework of GEP value transformation and the characteristics of ecological value.

## 2. Materials and Methods

### 2.1. Introduction of Study Area and Data Source

Hanwang town is located in the southwest corner of Tongshan District, Xuzhou City, in Jiangsu Province. There are 9 administrative villages in the town, covering an area of 63.93 square kilometers. It is adjacent to the Yunlong Lake Scenic Spot to the north, and across the lake from Xuzhou city. Hanwang town has a long history, as early as 205 BC when Chu and Han rivalries recorded here, whose name come from the war. There are two famous springs in Hanwang town, namely Bajian Spring and Maba Spring, related to the historical stories of Han Emperor Liu Bang. During the War of Liberation, many military and political organs of the General Front Committee of the Crossing River Campaign were located in Hanwang town. There are a network of rivers and mountains in Hanwang town, which has rich tourism resources. Yudai River connects the Beijing-Hangzhou Grand Canal in the north and Suihe in the south. In 2010, it was named "the most distinctive tourist Town in China", and in 2013, it was named as the national ecological town. Hanwang town has a good ecological foundation and is close to Xuzhou urbanized area. It has the potential to realize the transformation of ecological value and has typical significance of studying ecological towns. therefore, it was chosen as study area.

The data of land use status in Hanwang town are from the third National Land survey data in 2020 provided by Xuzhou Natural Resources and Planning Bureau. In order to ensure the timeliness of data update, the land use data was manually corrected and adjusted by comparing with updated Landsat 8 satellite images. The parts with blurred images and difficult to determine land use are manually corrected after field investigation by UAV. Other data (including price data) in this study were collected from Xuzhou Statistical Yearbook—2020 (total No. 33), Statistical Bulletin of Xuzhou National Economic and Social Development in 2020, Tongshan Yearbook 2020, and Three-year Data Analysis Report of Hanwang town, Xuzhou City, Jiangsu Province, Edition 2020, Hanwang administrative boundary data [39,40].

### 2.2. Classification of Ecosystem Types

The classification of ecological environment type is to facilitate the subsequent calculation of the functional and value of ecological environments. The ecological environment system is one of the most complex systems in nature. The common classification criteria of ecosystem types in China are farmland, forest land, grassland, water, wetland, settlement and desert. Hanwang town ecological types are divided into forest land, wood land, wetland, grass land, farmland, construction land and bare land. Figure 1 shows the area and spatial distribution of each ecological environment in Hanwang town.

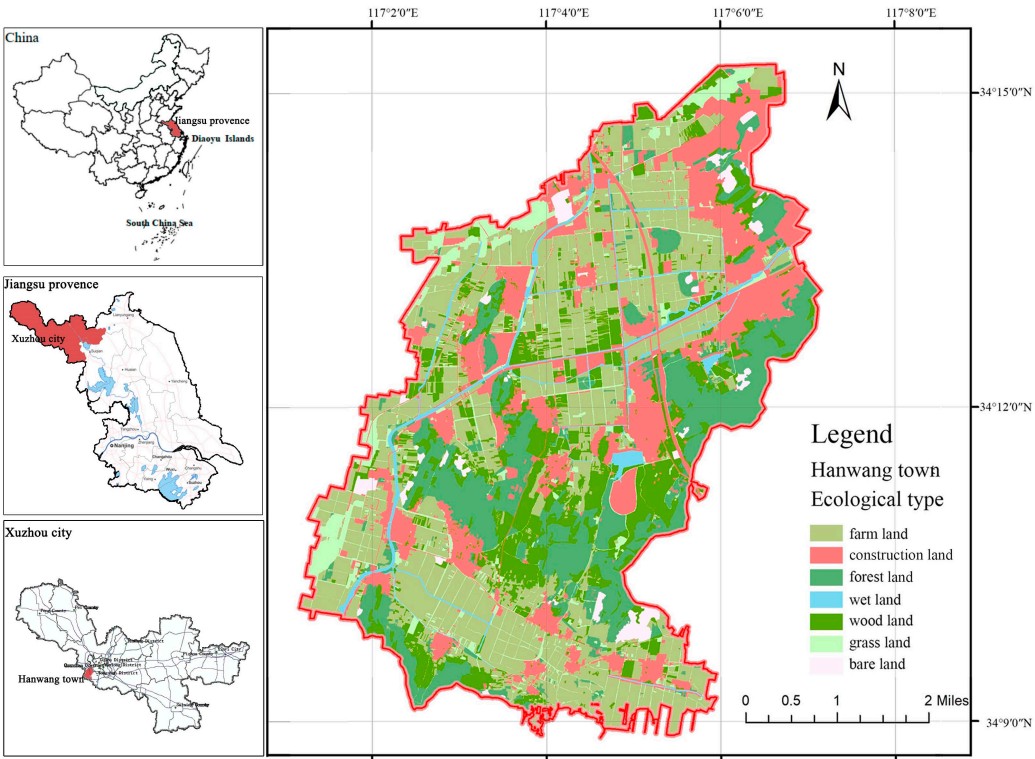

**Figure 1.** Ecological type distribution map.

*2.3. Construction of the GEP Accounting Index System*

2.3.1. Selection of Accounting Index

According to the characteristics of the ecology and environment system of the township, the index of ecological environment value accounting is selected through literature review summary. The GEP value specifically includes the value of product provision, the value of regulation services and the value of cultural tourism (Table 1). The product provision value includes agriculture, forestry, animal husbandry, fishery product supply value, water resource value and ecological energy value. The regulation service value includes soil conservation, water conservation, carbon sequestration and oxygen release, flood regulation and storage, air purification, water purification and climate regulation. The value of cultural tourism mainly refers to the total consumption price of tourist groups in the region.

2.3.2. Construction of the Accounting System

The GEP accounting index system of townships is composed of three indicators, which are product provision, regulation function and cultural tourism, and product provision is divided into six second-level indicators, namely agriculture, forestry, animal husbandry, fishing, water and energy. The regulation function is divided into seven secondary indexes: water conservation, soil and water conservation, carbon fixation and oxygen release, air purification, climate regulation, water purification, flood regulation and storage. Cultural tourism is mainly an indicator of natural landscape. The whole evaluation system is divided into 3 first-level indicators and 14 s-level indicators, as shown in Table 2.

**Table 1.** Ecological environment value accounting index and functional correlation table.

| Function Type | Accounting Content | Content Description | Forest | Bush Wood | Grass Land | Wet Land | Farm Land | Construction Land | Bare Land |
|---|---|---|---|---|---|---|---|---|---|
| Product provision | Agricultural products | Planting agricultural primary products, mainly including cereal, potato and other crop products | - | + | - | + | √ | - | - |
| | Forest products | Forest primary products, mainly wood, walnut, also include understory products | √ | √ | - | - | - | - | - |
| | Animal husbandry products | Livestock farming provides primary agricultural products such as eggs, milk, etc. | - | + | √ | - | - | + | + |
| | Fishery product | Aquatic products, including fish, shrimp and crabs | - | - | - | √ | - | - | - |
| | Water resources | Provide water resources that can be used directly, including water for production, life and ecology | √ | √ | √ | √ | - | - | - |
| | Ecological energy | Mineral energy and biomass energy as well as other wind energy, water energy, etc. | √ | √ | √ | + | √ | - | - |
| Regulation services | Water conservation | Block and intercept precipitation, enhance rainwater infiltration, conserve soil, replenish groundwater, and regulate surface runoff | √ | √ | √ | √ | √ | - | - |
| | Soil conservation | Keep water and soil, and reduce the erosion of soil by water, rain and wind | √ | √ | √ | √ | √ | - | - |
| | Flood control and storage | Save water and weaken the impact of flood peak transit | + | + | + | √ | + | - | - |
| | Carbon fixation and oxygen release | Plants absorb carbon dioxide and release oxygen, reducing the greenhouse effect | √ | √ | √ | √ | √ | - | + |
| | Air purification | Ecosystems block, absorb harmful dust and sulfur dioxide, nitrogen dioxide, clean the air | √ | √ | √ | √ | √ | - | - |
| | Water purification | The adsorption, decomposition, transformation and absorption of pollutants in the water to purify the water body | - | - | - | √ | - | - | - |
| | Climate regulation | Transpiration lowers temperature, alters local microclimates, increases humidity and lowers temperature | √ | √ | √ | √ | √ | - | - |
| Cultural tourism | Tourism consumption | The ornamental entertainment and health benefits of ecological environments bring consumers' travel consumption expenditure | √ | √ | √ | √ | √ | √ | + |

Note: √ means include ecological and environmental value function accounting; + represents the function of ecological and environmental value, but it is not included in the calculation; - indicates that this environment function is not available.

**Table 2.** Accounting system table.

| First Level Indicator | Second Level Indicator | Accounting Category | Value Quantity Index | Value Calculation Formula | |
|---|---|---|---|---|---|
| Product provision | Agricultural products | Food crop<br>Vegetable<br>Oil<br>Fruit | Food crop value<br>Vegetable value<br>Oil crop value<br>Fruit value | $V\mathrm{farm} = \sum\limits_{i=1}^{n} Ei \times P_i$ | [28] |
| | Forest products | Wood<br>Understory product | Timber value<br>Value of understory products | $V\mathrm{forest} = \sum\limits_{i=1}^{n} Ei \times P_i$ | |
| | Animal husbandry products | Livestock<br>Milk<br>Eggs<br>Fur<br>Other | Livestock value<br>Milk product value<br>Egg value<br>Wool value<br>Honey value | $V\mathrm{graziery} = \sum\limits_{i=1}^{n} Ei \times P_i$ | |
| | Fishery product | Fresh water product | Fresh water product value | $V\mathrm{fishing} = \sum\limits_{i=1}^{n} Ei \times P_i$ | |
| | Water resources | Mineral water | Water value | $V\mathrm{water} = \sum\limits_{i=1}^{n} Ei \times P_i$ | |
| | Ecological energy | Energy, biomass fuel, etc. | Ecological energy value | $V\mathrm{eco} = \sum\limits_{i=1}^{n} Ei \times P_i$ | [28] |

**Table 2.** *Cont.*

| First Level Indicator | Second Level Indicator | Accounting Category | Value Quantity Index | Value Calculation Formula | |
|---|---|---|---|---|---|
| Regulation services | Water conservation | | Water conservation value | $VQ = \sum_{i=1}^{j} (P_i - ET_i) A_i \times Y_i$ | [41] |
| | Soil conservation | | Reduce sediment value Reduce source pollution value | $V_S = R \times K \times L \times S \times (1 - C) \times S_q \times P$ | [28] |
| | Flood control and storage | | Value of flood control and storage | $C_f = (C_l + C_r)P = (e^{4.904} \times A^{0.927} \times T + C_t \times 0.35)P$ | [24] |
| | Carbon fixation and oxygen release | | Carbon fixation value | $V_c = 162 \times \sum_{j=1}^{n} N_{pj} \times A_j \times P$ | [41] |
| | | | Oxygen release value | $V_o = 120 \times \sum_{j=1}^{n} N_{pj} \times A_j \times P$ | |
| | Air purification | | Purifying sulfur dioxide value Purifying nitrogen oxide value Dust purification value | $V_a = \sum_{i=1}^{m} \sum_{j=1}^{n} Q_{ij} \times S_i \times P$ | [28] |
| | Water purification | | The consumption value of COD treatment Nitrogen consumption value Phosphorus consumption value | $V_{wp} = \sum_{i=1}^{n} Q_i \times A \times P$ | [28] |
| | Climate regulation | | Cooling value Humidification value | $V_x = (G_a \times H_a + W_a \times E_p \times \beta) \times \rho \times P$ | [42] |
| Cultural tourism | Revenue from tourism | | Landscape recreation value | $V\text{tourism} = V_s + V_t = V_a \times T \times r + V_b \times T,$ $v = \dfrac{V\text{tourism}}{\sum_{j=1}^{n} S_j \times T_j \times W_j}$ | [43,44] |

## 3. Results

According to the accounting, the total GEP of Hanwang town is 1.165 billion CNY, among which the product supply value is 106 million CNY, the adjustment service value is 239 million CNY, and the cultural tourism value is 820 million CNY. The distribution of GEP in township is not balanced, whose feature can be shown as Figure 2. The area is composed of Yudai River with and its nearby tourist attractions; "Three Areas" refers to the three areas with high cultural tourism value.

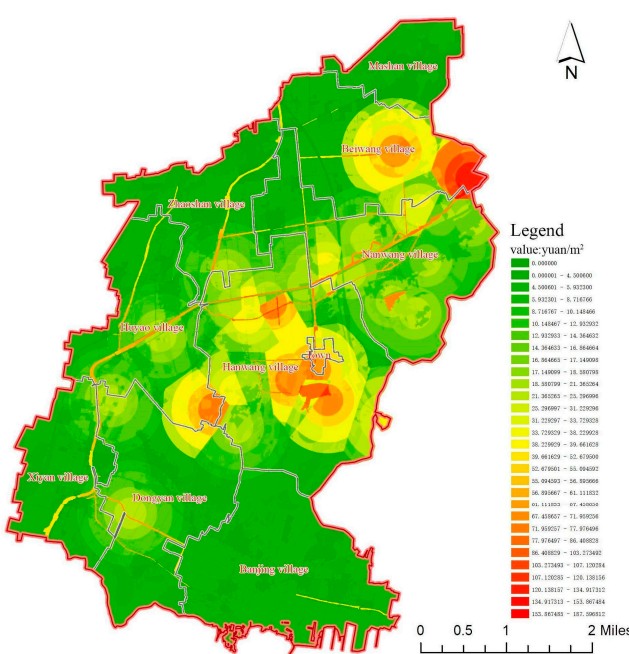

**Figure 2.** GEP value distribution map.

### 3.1. Product Value and Its Spatial Distribution Characteristics

The total value of agricultural, forestry, animal husbandry and fishery products in Hanwang town is 96,021,310 CNY (Table 3). In terms of land use type, farm land provides the highest value, with a unit price of 2.2502 CNY per square meter. Followed by wetland, the unit price reached 1.8482 CNY per square meter. The unit area value of forest land is 0.1044 CNY per square meter, and the unit value of woodland reaches 0.0887 CNY per square meter. From the perspective of spatial, the overall distribution features are high in the east but low in the west (Figure 3a). The areas with high value of agricultural products are mainly distributed in Banjing Village and Dongyan Village (Figure 4a). The areas with high value of forestry products were mainly distributed in Hanwang Village and Nanwang Village (Figure 4b). The value of animal husbandry products presents the spatial characteristics of high in the west but low in the east (Figure 4c). The value distribution of fishery products showed a spatial feature of high value in the middle but lower value in the north and south (Figure 4d).

**Table 3.** Table of total value of Product provision.

| Village | Agricultural Products (CNY) | Forestry Products (CNY) | Animal Husbandry Products (CNY) | Fishery Products (CNY) | Water Resources Supply Value (CNY) | Ecological Energy Supply Value (CNY) |
|---|---|---|---|---|---|---|
| Banjing | 18,849,745.63 | 184,665.78 | 47,980.91 | 95,761.76 | 304,996.99 | 878,278.98 |
| Beiwang | 10,998,834.56 | 114,352.22 | 29,520.43 | 228,755.72 | 271,240.58 | 529,763.06 |
| Dongyan | 10,776,146.15 | 297,608.15 | 44,152.58 | 213,858.93 | 491,480.80 | 966,305.16 |
| Hanwang | 10,273,522.47 | 433,971.77 | 48,280.12 | 684,282.92 | 883,368.37 | 1,272,506.12 |
| Town center | 0.00 | 238.46 | 1239.17 | 7302.60 | 4849.25 | 2378.26 |
| Huyao | 10,737,137.78 | 165,189.55 | 50,366.91 | 398,160.79 | 430,374.57 | 671,311.51 |
| Mashan | 1,930,124.93 | 48,646.66 | 46,278.27 | 0.00 | 94,793.93 | 218,836.91 |
| Nanwang | 10,047,122.78 | 281,893.32 | 28,103.85 | 759,426.75 | 728,369.90 | 894,876.33 |
| Xiyan | 8,474,000.93 | 65,268.19 | 79,783.53 | 340,921.56 | 308,746.75 | 440,210.32 |
| Zhaoshan | 8,769,673.30 | 35,149.07 | 98,246.16 | 335,595.06 | 285,075.10 | 403,733.91 |
| Total | 90,856,308.53 | 1,626,983.17 | 473,951.93 | 3,064,066.09 | 3,803,296.24 | 6,278,200.56 |

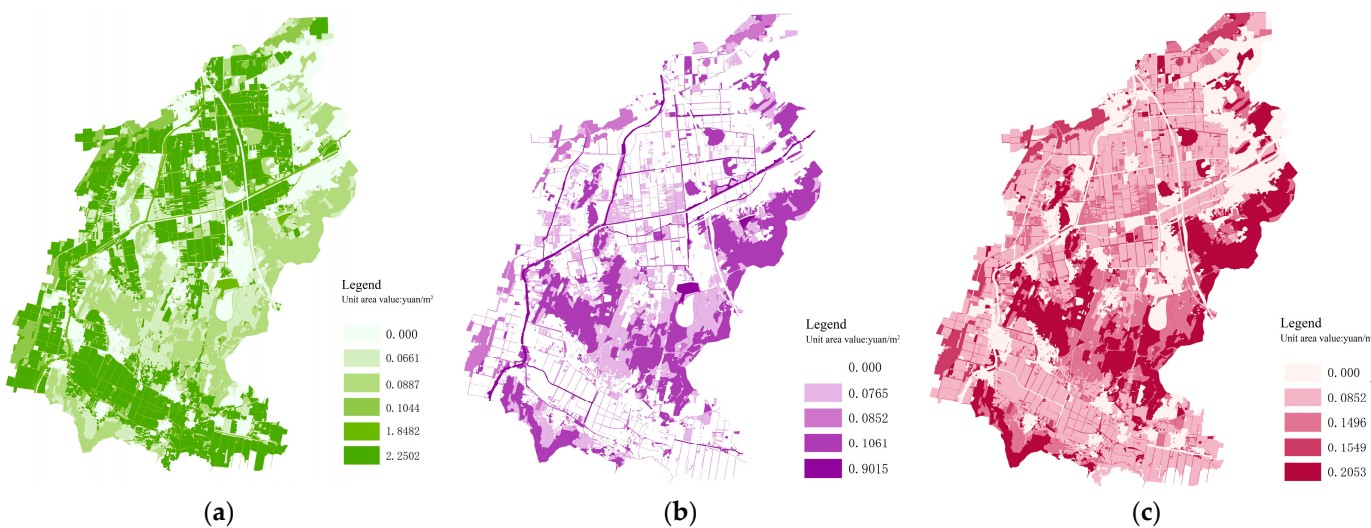

**Figure 3.** Chart of supply value per unit area. (**a**) Agriculture, forestry, animal husbandry and fishery; (**b**) water resources; (**c**) ecological energy.

The total water supply value of Hanwang town is 3,803,300 CNY. In terms of ecology types, the water supply capacity of wetlands is the strongest (Figure 3b), reaching 0.9015 CNY/square meter. The main rivers are Yudai River and Xia River, which flow from south to north. The water supply capacity of forest is the second strongest, whose value

can reach 0.1061 CNY/square meter. The water supply value of grassland is low, and the unit value can reach 0.0852 CNY/square meter. The value of woodland is even lower than grassland, reaching 0.0765 CNY/m². Farmland requires irrigation, so the value of the water supply will be offset, which is not included in this article. Dongyan Village and Hanwang Village have the highest water resource value (see Figure 4e).

The total value of ecological energy supply in Hanwang town is 6,278,200 CNY (Table 3). From the perspective of ecological type, the strongest supply capacity belongs to the forest land, whose unit value can reach 0.2053 CNY per square meter (Figure 3c). Grassland followed, reaching 0.1529 CNY per square meter. The supply capacity of shrub is 0.1496 CNY per square meter, which is close to that of grassland. The ecological energy supply value is high in the central part but low in the west and north of the town (Figure 3c). Ecological energy supply capacity is related to the biomass provided by the natural environment. Hanwang Village has the most biomass resources because there are Ma Mountain, Donkey Yan Mountain and Guangshan Mountain. That is why its ecological energy supply value is the highest (Figure 4f).

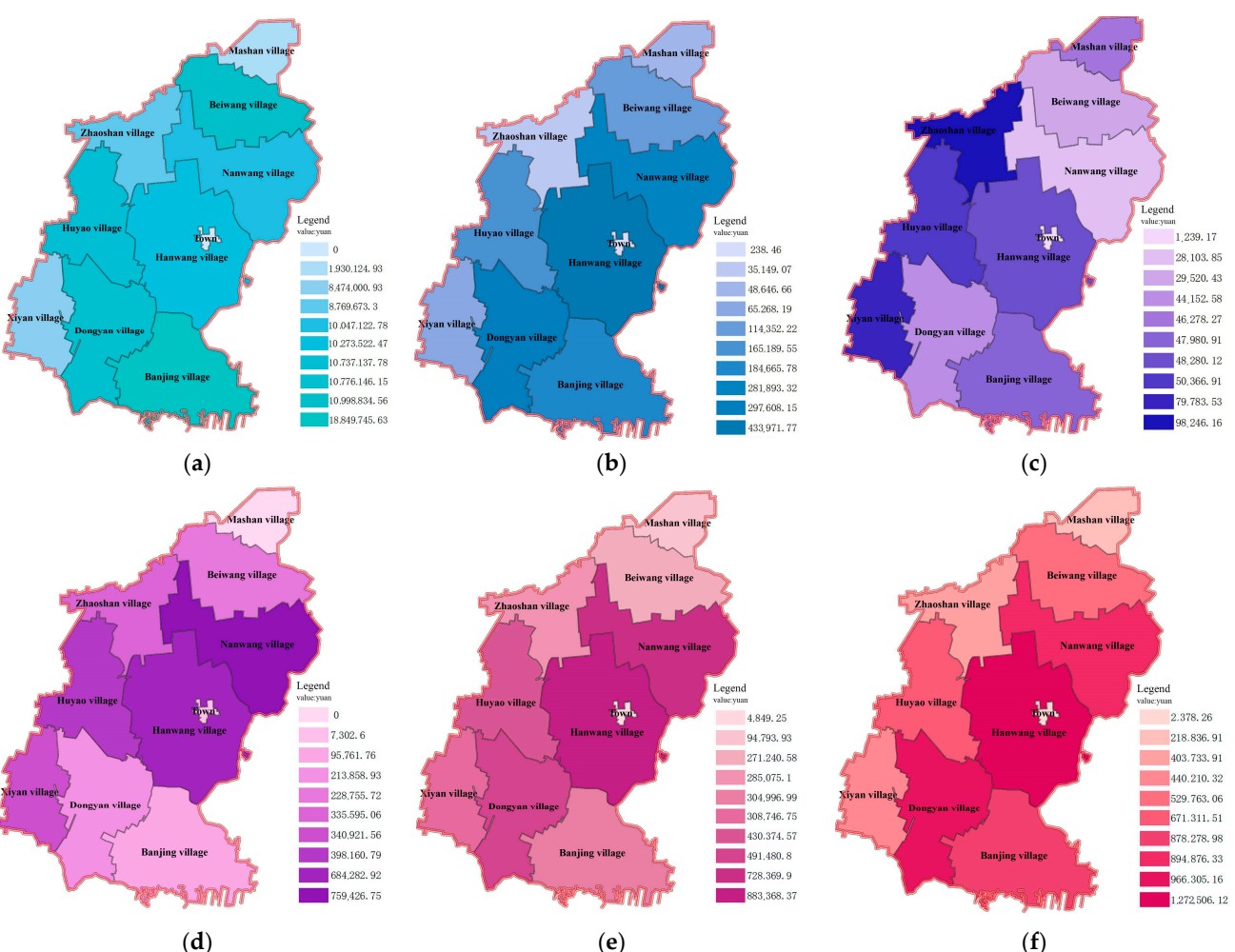

**Figure 4.** Supply value map in each village. (**a**) Agriculture; (**b**) forestry; (**c**) animal husbandry; (**d**) fishery; (**e**) water resources; (**f**) ecological energy.

## 3.2. Regulation Services Value and Its Spatial Distribution Characteristics

The regulation services value of Hanwang town is 238,833,900 CNY (Table 4), which is about 2.5 times of the product provision value. Figure 5a shows the distribution characteristics of regulation services value. From the perspective of ecological type, wetland had the highest value, reaching 49.9298 CNY/square meter. The next is forest land, whose value is 5.5084 CNY/square meter. Then, the woodland reached 4.1735 CNY per square meter; The

value of grassland regulation service is low, which is 4.1561 CNY/square meter. Lowest is farmland, whose value is only 1.6429 CNY/square meter. The distribution of features regulation service value is high in the middle but low in the north (Figure 6a). Hanwang Village has a large area of wetland and forest, so the regulation services value is the highest, reaching 51,611,200 CNY. Nanwang Village also has a high value due to the distribution of Shigang Reservoir, Yudai River and Lali Mountain.

**Table 4.** Table of regulation services value in Hanwang town.

| Village | Water Conservation Value (CNY) | Soil Conservation Value (CNY) | Flood Storage Value (CNY) | Carbon Fixation and Oxygen Release Value (CNY) | Air Purification Value (CNY) | Water Purification Value (CNY) | Climate Regulation Value (CNY) |
|---|---|---|---|---|---|---|---|
| Banjing village | 6,261,533.59 | 1,810,564.31 | 1,844,027.79 | 3,066,552.47 | 3,629,811.14 | 100,098.55 | 5,621,664.62 |
| Beiwang village | 4,571,374.51 | 3,563,233.41 | 4,405,014.37 | 1,908,386.95 | 2,286,547.14 | 239,115.45 | 3,566,117.88 |
| Dongyan village | 7,695,825.52 | 5,013,324.99 | 4,118,155.62 | 3,338,226.54 | 3,550,368.93 | 223,544.03 | 7,868,524.20 |
| Hanwang village | 12,171,030.36 | 13,166.05 | 13,176,833.55 | 4,508,787.71 | 4,749,978.34 | 715,272.25 | 11,275,956.21 |
| Town center | 49,998.34 | 2,468,264.92 | 140,621.83 | 10,807.23 | 12,177.38 | 7633.31 | 27,264.72 |
| Huyao village | 6,388,569.60 | 819,535.23 | 7,667,147.98 | 2,438,086.08 | 2,808,819.21 | 416,192.41 | 5,096,540.33 |
| Mashan village | 1,528,921.21 | 3,535,230.44 | 0.00 | 746,475.96 | 752,892.59 | 0.00 | 1,800,838.75 |
| Nanwang village | 9,562,563.10 | 1,616,939.77 | 14,623,834.06 | 3,288,746.82 | 3,658,532.92 | 793,819.14 | 7,662,788.41 |
| Xiyan village | 4,475,464.94 | 1,465,124.06 | 6,564,925.86 | 1,652,445.00 | 1,978,577.53 | 356,360.98 | 3,125,669.03 |
| Zhaoshan village | 4,201,034.83 | 2,315,992.42 | 6,462,356.71 | 1,538,850.82 | 1,889,710.71 | 350,793.26 | 2,701,728.84 |
| Total | 56,906,316 | 22,621,375.6 | 59,002,917.77 | 22,497,365.58 | 25,317,415.89 | 3,202,829.38 | 48,747,092.99 |

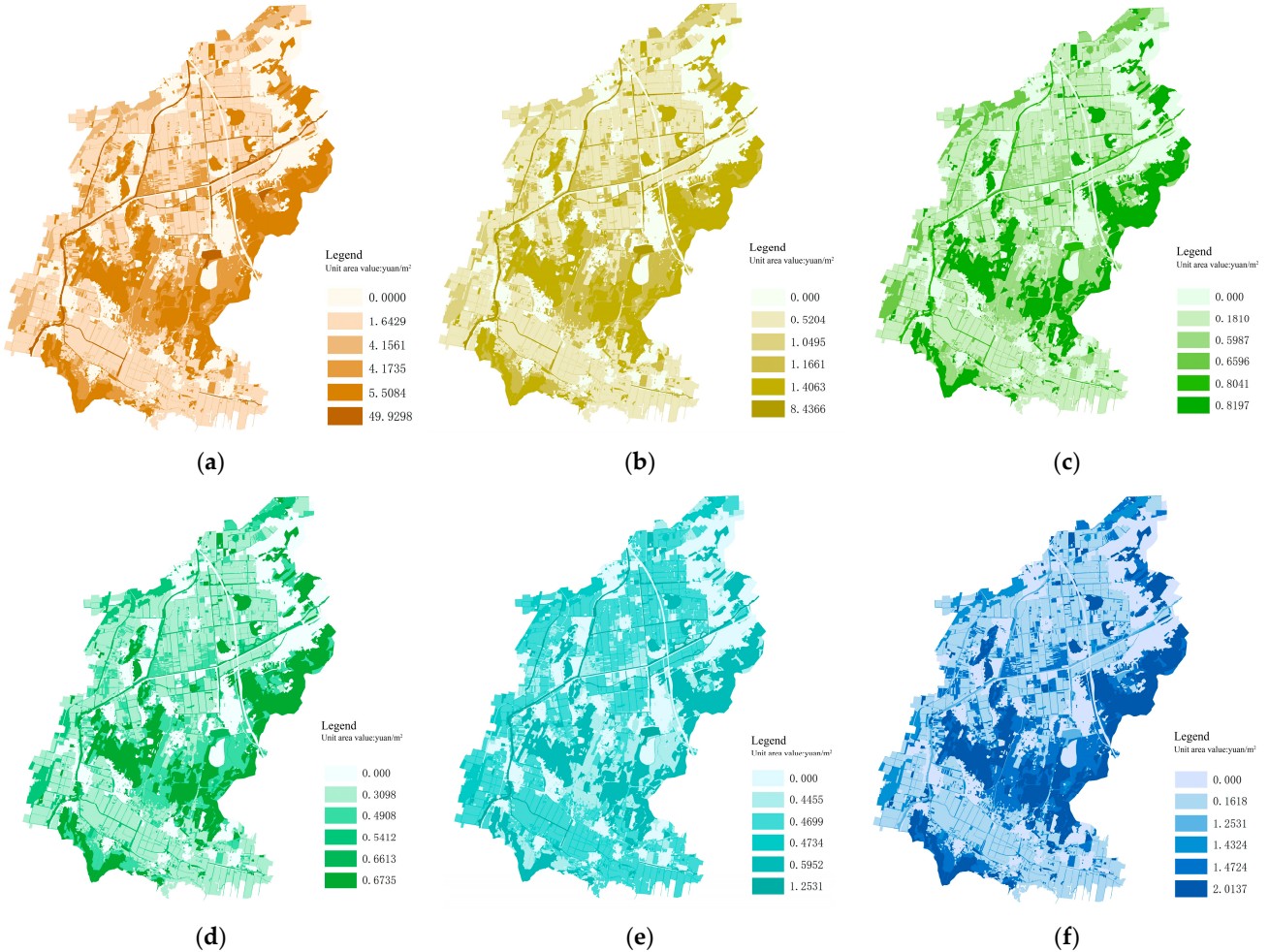

**Figure 5.** Adjusted value distribution map. (**a**) Comprehensive adjustment value distribution map Water conservation value distribution map; (**b**) water conservation; (**c**) soil conservation; (**d**) carbon fixation and oxygen release; (**e**) air purification; (**f**) climate regulation.

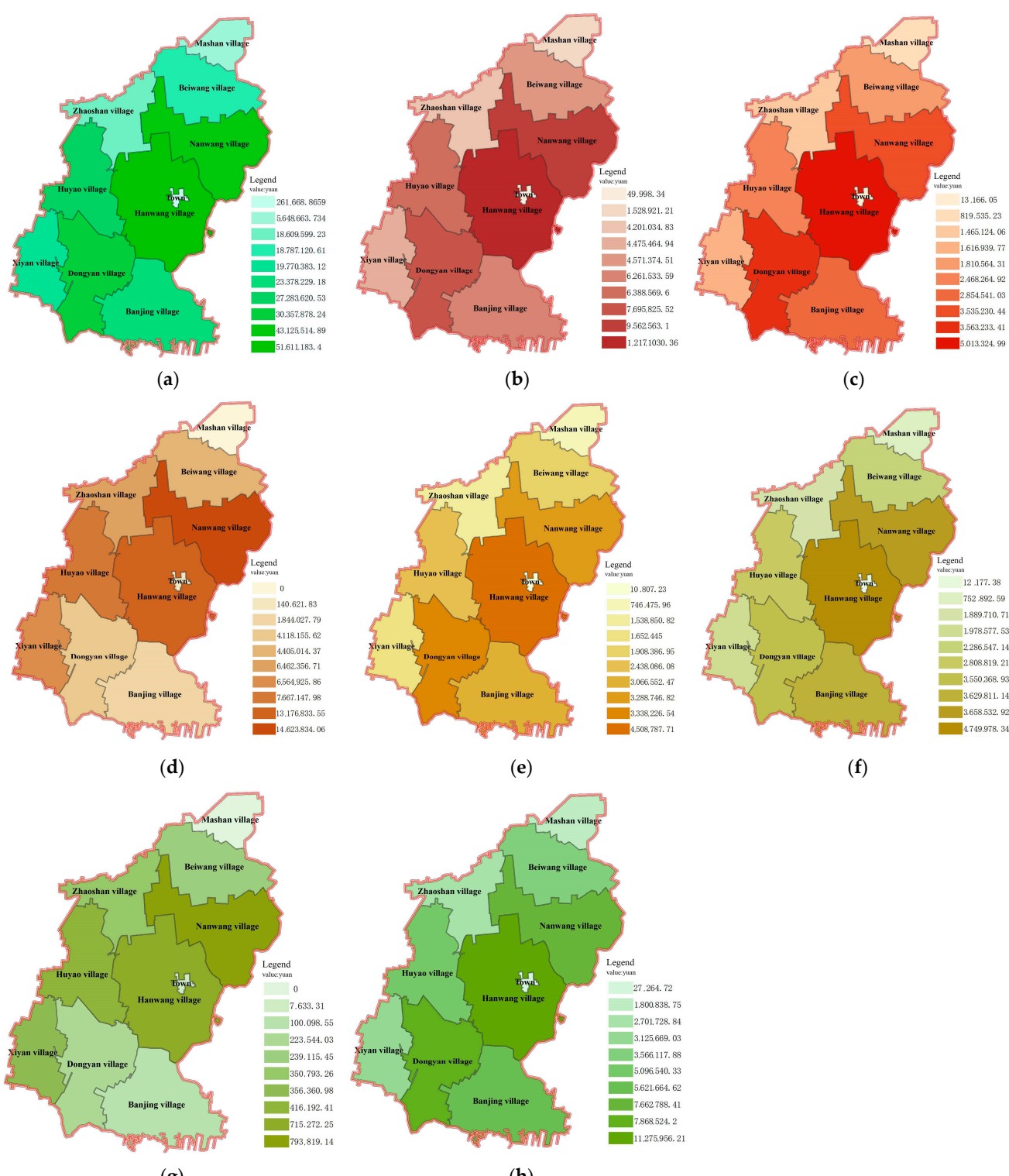

**Figure 6.** Adjusted value distribution map in each village. (**a**) Comprehensive adjustment value distribution map; (**b**) water conservation; (**c**) soil conservation; (**d**) water control and storage; (**e**) carbon fixation and oxygen release; (**f**) air purification; (**g**) water purification; (**h**) climate regulation.

The water conservation value of Hanwang town is 56,906,316 CNY (Table 4). The strongest regulation services ability belongs to wetland (Figure 5a), because the water conservation and regulation value of lakes and rivers in plain area is the highest (Figure 5b). The soil conservation value within Hanwang town is 22,621,375 CNY, and the forest has the strongest soil conservation ability (Figure 5c). The distribution characteristics is high

in southeast but low in northwest (Figure 6a). The value of carbon sequestration and oxygen release in Hanwang town is 22,497,400 CNY. The overall distribution is high in the southeast but low in the northwest (Figure 5d). Hanwang Village, Nanwang Village and Banjing village have a much higher value (Figure 6d) in carbon sequestration and oxygen release value because of higher forest coverage.

The total value of air purification in Hanwang town is 25,317,415 CNY. The two ecological types that have the greatest impact on air purification are wetland and forest land. Villages with large reservoirs and forest land have more value in air cleaning (Figure 5e). The total value of climate regulation within Hanwang town is 48,747,192 CNY, and the woodland has the strongest climate regulation ability (Figure 5f). The level of climate regulation ability is related to green quantity and evaporation, especially the green quantity of green broad-leaved forest, which provides the strongest regulation ability. The evaporation capacity of the reservoir is large, and it can also provide climate regulation value.

The water purification value in Hanwang town is 3,202,829 CNY, and the flood storage value is 59,002,917 million CNY. The two values have strong relationship with the water storage capacity of rivers, lakes and reservoirs. Nanwang Village, Hanwang Village and Huyao Village have higher flood regulation and storage value because of the large water within the village (Figure 6e). From the distribution feature of water purification value of each village in the town, the situation is high value in the middle but low value in the north. Nanwang Village and Hanwang Village have the highest water purification value (Figure 6g) because there are reservoirs and rivers. Huyao Village and Dongyan Village also have high water purification value due to Yudai River and Xixia River flowing through them.

### 3.3. Cultural Tourism Value and Its Spatial Distribution Characteristics

According to the calculation, the annual value created by tourism in Hanwang town is 820 million CNY. According to the research on the spatial distribution of tourism value, the cultural tourism value per unit area can be backderived from the perspective of tourism resource distribution space, so as to form the spatial visualization analysis result of value [44]. The spatial distribution of cultural tourism value should be affected by scenic spot grade and spatial distance. Therefore, tourist attractions can be divided into four levels according to their importance. The first level has one scenic spot named Xuzhou Paradise, the second level has 6 scenic spots, the third level has 7 scenic spots, and the fourth level has 18 scenic spots. The index of importance of first-class scenic spots is 8, that of second-class scenic spots is 4, that of third-class scenic spots is 2 and that of fourth-class scenic spots is 1. Each scenic spot is divided into a single scenic spot value weight according to the distance level of 300 m, 500 m, 800 m and 1000 m. The specific standard is that the weight within 300 m is 40%, the weight between 300 m and 500 m is 30%, the weight between 500 m and 800 m is 20%, and the weight between 800 m and 1000 m is 10%.

According to the spatial distribution characteristics of the cultural tourism value of Hanwang town (Figure 7a), the areas with high value are mainly distributed in the north and middle areas of the town. The core area of Xuzhou Paradise with the highest unit area value reaches 134.91 CNY/square meter. The unit area value of the core area of the former site of the general Front Committee of the River Crossing Campaign has reached 67.46 CNY/square meter. The value of cultural tourism in the town center area is relatively high, which have formed three distinct clusters. It is find that the three clusters have a certain corresponding relationship with the results of nuclear density analysis about tourism attraction. The cultural tourism value per unit area of the core area of Yueliang Bay Ecological Farm in Group 1 reached 67.46 CNY/square meter, and the core areas of Hanwang Tourist Scenic Spot, Zishan Mountain and Fengling Mountain Scenic spot in Group 2 also reached 67.46 CNY/square meter. The cultural tourism value per unit area of Pana Valley core area in Group 3 also reached 67.46 CNY per square meter.

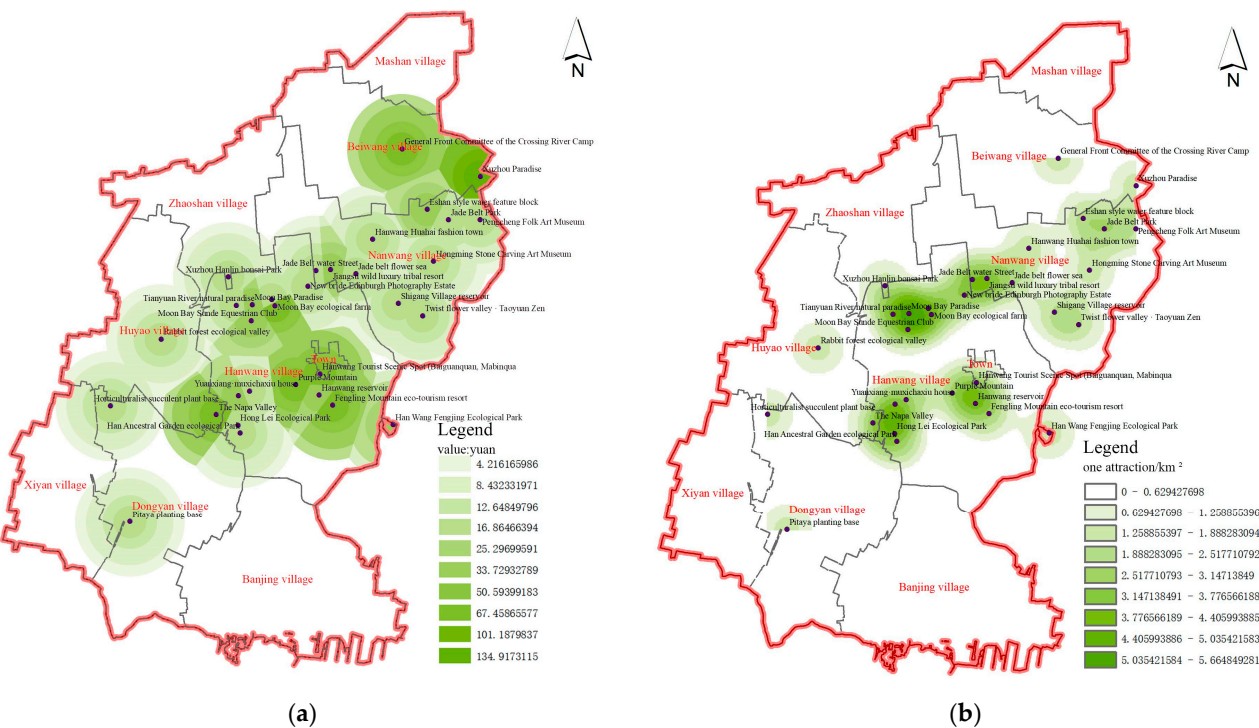

**Figure 7.** Attractions and value distribution. (**a**) Cultural tourism value distribution map; (**b**) Nuclear density map of tourist attractions.

From the distribution density of tourist attractions in Hanwang town (as shown in Figure 7b), the higher area is mainly distributed in the part of the town, namely Hanwang Village (including land directly under the town), and there are three groups with higher density in the village. The average nuclear density of the three clusters is about 5 points per square kilometer; Secondly, the overall density area is located in the northeast direction, mainly in Nanwang Village. The landscape distribution in other areas is scattered, representative of which are Xuzhou Paradise in Beiwang Village, the former site of the General Front Committee of the Yangtze River Crossing Campaign, Wang Fengjing Ecological Park of Han Dynasty, and dragon fruit planting base in Dongyan Village.

## 4. Industry Strategy Discussion

There are four main modes to transform the ecological value of the town, which are ecological industry management, ecological value promotion, ecological resource index trading and ecotourism (Table 5). Though ecological value promotion can improve GEP accounting result, it is difficult to translate into economic benefits because of the public property of this product. The difference is that the other three kinds are easier to realize the transformation of ecological value. It is found that the GEP accounting value of wetland, forest and tourism areas is significantly higher than that of other areas. Therefore, Hanwang town should strengthen the protection of wetlands and forests, and expand tourist attractions and projects as far as possible without damaging the environment. The study suggests that the value of GEP can be increased by first identifying areas with low food production values and then growing high value-added foods. There are many forests in the town, but lacking of understory planting industry, which is needed to develop. Hanwang town has a number of coal mining collapsed abandoned ponds, which can be developed into fish farming areas. At present, the two springs can be developed into commercial mineral water, and the ecological value is huge. The value of carbon fixation, oxygen release, air purification and water purification can be sold through the established emissions trading platform to promote the transformation of ecological value. The ecological aesthetic value is mainly manifested through the income of tourism. It is

required to expand new scenic spots for further improvement of the value including the combination of management about abandoned coal mining areas and rotten villa area. In addition, service facilities such as parking, accommodation and shopping should be increased to promote consumption and increase tourism income.

**Table 5.** Table of measures for ecological value transformation.

| First Level Indicator | Second Level Indicator | Concrete Measures | Four Modes for Value Transformation | Easy or Not |
|---|---|---|---|---|
| Product provision | Agricultural products | High value-added food can be grown, such as organic rice. | Ecological industry management | √ |
| | Forest products | Understory industry needs to be explored. | | |
| | Animal husbandry products | Increase the special breeding industry. | | |
| | Fishery product | Coal mining collapsed abandoned ponds can be developed into fish farming areas. | | |
| | Water resources | The two springs can be developed into commercial mineral water. | | |
| Regulation services | Ecological energy | The existing bare land of coal and stone mining in the town can be fixed by ecological restoration. By increasing the extent of vegetation coverage, the value will be improved. | Ecological value promotion | - |
| | Water conservation | | | |
| | Soil conservation | | | |
| | Flood control and storage | | | |
| | Carbon fixation and oxygen release | Building a carbon emission trading platform. | Ecological resource index trading | √ |
| | Air purification | Building a emissions trading platform. | | |
| | Water purification | | | |
| | Climate regulation | By changing the ecological type of the surface through afforestation | Ecological value promotion | - |
| Cultural tourism | Tourism consumption | It requires the expansion of new scenic spots. In addition, service facilities should be increased to promote consumption and increase tourism income. | Ecotourism | √ |

Note: √ means that such measures are easy to transform ecological value into industry GDP.

*4.1. Guiding Strategies for Agricultural Development*

4.1.1. Improve the Quality and Added Value of Agriculture

The main agricultural type of Hanwang town is field crop cultivation, which mainly produces wheat, rice, corn and other food crops. Its economic value is estimated to be as high as 2.25 CNY/square meter. It is suggested to replace the varieties with high economic value, good taste quality and suitable for planting. Hanwang needs to experiment with organic farming, such as the rice-crab symbiosis model to increase the value of products because organic products can be ten times that of non-organic foods. In addition, the market in Xuzhou has a great demand for fruits, vegetables and flowers. Hanwang town have unique geographical advantages, whose fruits, vegetables and flowers can realize the logistics from picking to consumption within one hour. According to the fruit tree planting plan, the irrigated land near the village construction area can be transformed into fruit, vegetable and flower planting base to improve the industrial income (Figure 8a). Hanwang town is so rich in forestry resources that every village should develop a seedling industrial park, which combines landscape design, nursery stock research, planting, display and sales, and sightseeing experience (Figure 8c).

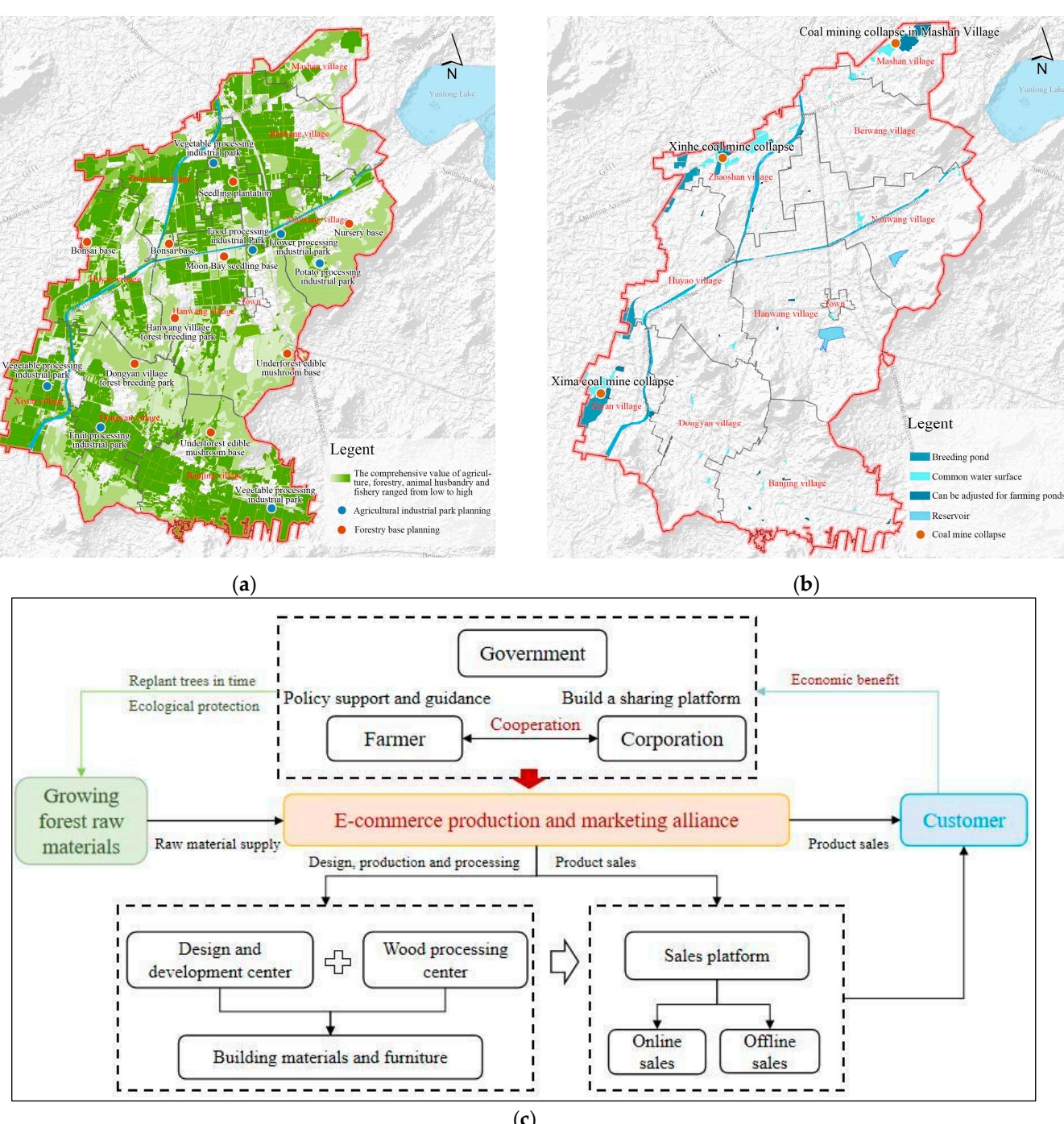

**Figure 8.** Agriculture, forestry, animal husbandry and fishery industry planning. (**a**) Plan of Hanwang town Agricultural Processing Industrial Park; (**b**) fishery spatial planning; (**c**) forestry business model diagram.

### 4.1.2. Development of Water Resources Industry

Hanwang town has high quality mineral water. The Bajian Spring in Hanwang town has a daily water output of 2000–2500 cubic meters. According to the conventional experience, 10% of the flow is reserved as the ecological base flow [45], so the daily development flow of Hanwang town is 1800 cubic meters−2250 cubic meters. Considering that Tujianquan plays a very important role in the ecological replenishment of Yudai River and Yunlong Lake, only 10% of it will be used as mineral water resources. At least 400,000–500,000 bottles of mineral water can be supplied to the market every day, if each

bottle is filled with 500 mL. The industry will generate about 300 million CNY a year. Considering the water output of Mabai Spring is similar to Bajian Spring, the two springs could generate 600 million CNY for Hanwang town annually.

The main aquaculture waters areas in Hanwang town are Hanwang Reservoir, Shigang reservoir, Xinhe coal mining collapse fish pond. The added value of fishery should be considered from two aspects: expanding the breeding area and increasing the additional income of products. In combination with the repair of Xinhe coal mine subsidence area, Hanwang town can expand the aquaculture water area and form a fishery aquaculture concentration area (Figure 8b). Xuzhou city has the tradition of koi culture, as early as the Eastern Han Dynasty, there are koi culture ornamental records. Considering that the global ornamental fish market demand is strong, Hanwang town should develop the koi culture industry. In addition, tourism can be combined with the development of recreational fisheries. Hanwang town can open free fishing areas in natural waters such as Yudai River and Xiahe River. Free fishing is an important way to attract tourists to Hanwang town, which can promote the coordinated development of tourism and recreational fishing.

*4.2. Guiding Strategy of Ecological Service Industry*

4.2.1. Ecological Restoration and Management

Hanwang town has a long history of mining, forming a large number of quarry pits that faced with ecological problems such as mountain destruction, soil fragmentation and forest destruction. However, they also have huge ecological value potential and should be regarded as potential industrial mines. Because reasonable development and utilization can make them become a new landmark to show the city's personality and characteristics. In order to better protect and utilize the industrial and mining heritage of Xinhe Coal mine, it is urgent to make restoration for underground goaf. After artificial restoration, landscape planning and design can be used to create an ecosystem with unique collapse wetland landform. South of the Hanwang Reservoir has a rotten end villa site, and a large area of bare wasteland. Plant engineering construction can improve the ecological regulation ability and increase the regulation service value. After professional evaluation, buildings can be used need reconstruction while cannot be used should be demolished to restore the ecological space.

4.2.2. Emissions Trading

(1)    Construct township carbon emission trading pilot project

The construction of township carbon emission trading pilot project needs government leadership, because it can ensure the perfection and credit of the trading platform. Secondly, we should pay attention to market-oriented trading principles and price transparency, so as to encourage enterprises in demand to purchase carbon emission trading rights. At present, the domestic carbon emission trading market towards the national trading started. In 2021, Xuzhou has started to implement the "Carbon Emission Trading Management Measures (Trial)", so, Hanwang town should actively integrate into the platform. It has been calculated that the total value of carbon fixation and oxygen release in Hanwang town is 22,497,365.58 CNY, among which the carbon fixation value is 129,244,018.52 CNY. This part of carbon sequestration value can be traded with traditional energy consumption enterprises such as steel, electric power and cement within Xuzhou City through Xuzhou Carbon Emission Trading Operation and Management Center. Thus, the ecological value of Hanwang town is transformed into economic value, and the value of ecological products is transformed.

(2)    Establish pilot projects for emission trading

Emission trading is a kind of "market incentive" environmental regulation. Relevant research shows that the implementation of emission trading has obvious effect on reducing pollution [46]. In 2015, Xuzhou Municipal Government promulgated <the Administrative Measures for Paid Use and Trading of Major Pollutant Emission Rights in Xuzhou (Trial)>,

which mainly provides guidance for pollutant emission monitoring, pollutant trading and trading management. It has been calculated that the total value of pollutants purification in Hanwang town is 28,520,245.25 CNY, among which the air purification value is 25,317,415.87 CNY, and the water purification value is 3,202,829.38 CNY. This part of the value can be traded with pollution production enterprises within the scope of Xuzhou City through the platform to obtain ecological value income.

### 4.3. Cultural and Tourism Industry Guidance Strategy

The tourism industry of Hanwang town needs to fully consider the distribution of resource value, landscape, road network construction and other conditions. The spatial structure of tourism is determined as "one core, three belts and four districts" (Figure 9a). In terms of spatial layout mode, Gunn's "community-attraction tourism spatial layout mode" is adopted [47], means that the tourism services distributed in the center of the tourist area and tourist attractions can be connected by tourist transportation lines. This model is widely used because it takes into account the interests of tourists and facilitates the holding of sightseeing activities.

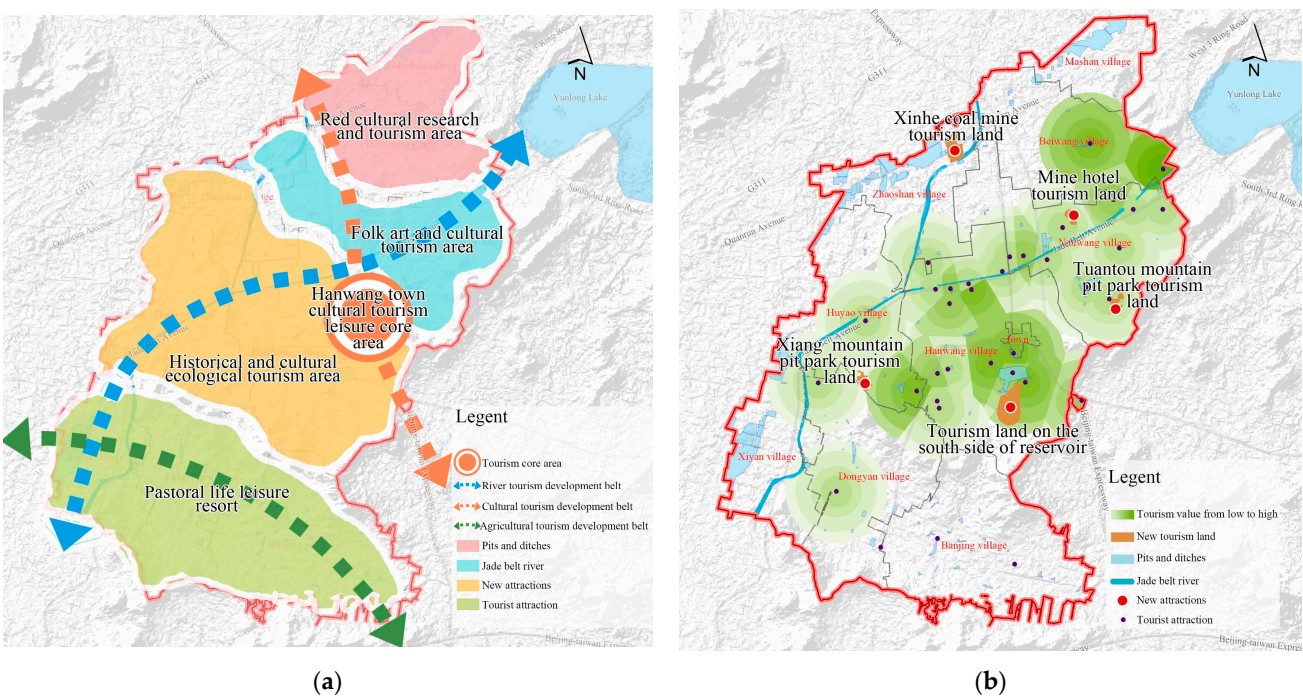

(**a**)　　　　　　　　　　　　　　　　　　　(**b**)

**Figure 9.** Tourism structure and land use adjustment. (**a**) Planning map of tourism space structure; (**b**) land use adjustment and new attractions.

From the perspective of spatial characteristics of regulation service value, the areas with the lowest value are mainly concentrated in construction land and wasteland, which can be adjusted into industrial land for tourism (Figure 9b). At present, there are homestays and villas in the mountains south of Hanwang Reservoir, but they are abandoned due to lack of management and maintenance. This causes serious waste of land, which can be adjusted into tourism construction land to provide construction space for the development of Hanwang tourism.

### 4.3.1. New Scenic Spots and Route Organization

First of all, we need to determine new types of special tourism products. At present, Hanwang town mainly develops six tourism products, including recuperation vacation tourism products, ecological tourism products, forest tourism products, river tourism products, sightseeing agricultural tourism products, historical and cultural tourism products. According to the existing resources, Hanwang town can also develop research tourism

products. Among them, the study tourism products are in the form of visits to relevant historical and cultural attractions, museums, exhibition halls or research and education bases. Xinhe Coal Mine in Hanwang town has abundant coal industry cultural relics, which should be protected and used for tourist visiting.

In general, Yudai Avenue and its extension road should be built in Hanwang town as the main line of town tourism, which is used to connect the surrounding sightseeing routes and scenic spots in the town. Other routes can be planned and designed on the basis of branch roads to create high-quality culture-themed tourism routes by improving the quality of roads and facilities. The route planning inside the scenic spot needs to use motor and non-motor lanes to connect various scenic spots in series. The length of the tour route should be appropriate, so as to avoid the tour fatigue of walking or cycling.

According to the characteristics of tourism resources type to determine the theme of tourism routes. According to the distribution characteristics of the existing cultural, ecological and agricultural resources in Hanwang town, three types of theme tourism routes in Hanwang town are determined, which are red research tourism routes, ecological cultural tourism routes, and pastoral sightseeing tourism routes (as shown in Figure 10a). The Red research tourism route is mainly connected with the Red Education base, the former site of the general Front Committee of the Crossing River Campaign in the north of the town, and the Xinhe Coal Mine industrial culture research base. The ecological humanistic tourism route mainly connects most ecological and historical humanistic landscape nodes in the town, such as Hanwang Tourist Scenic Spot and Pana Valley. The rural sightseeing route mainly connects the large agricultural space in the west and south. Based on the advantages of agricultural resources and good scenery, Hanwang town can develop rural sightseeing, ecological picking, parent–child farms and other agricultural projects.

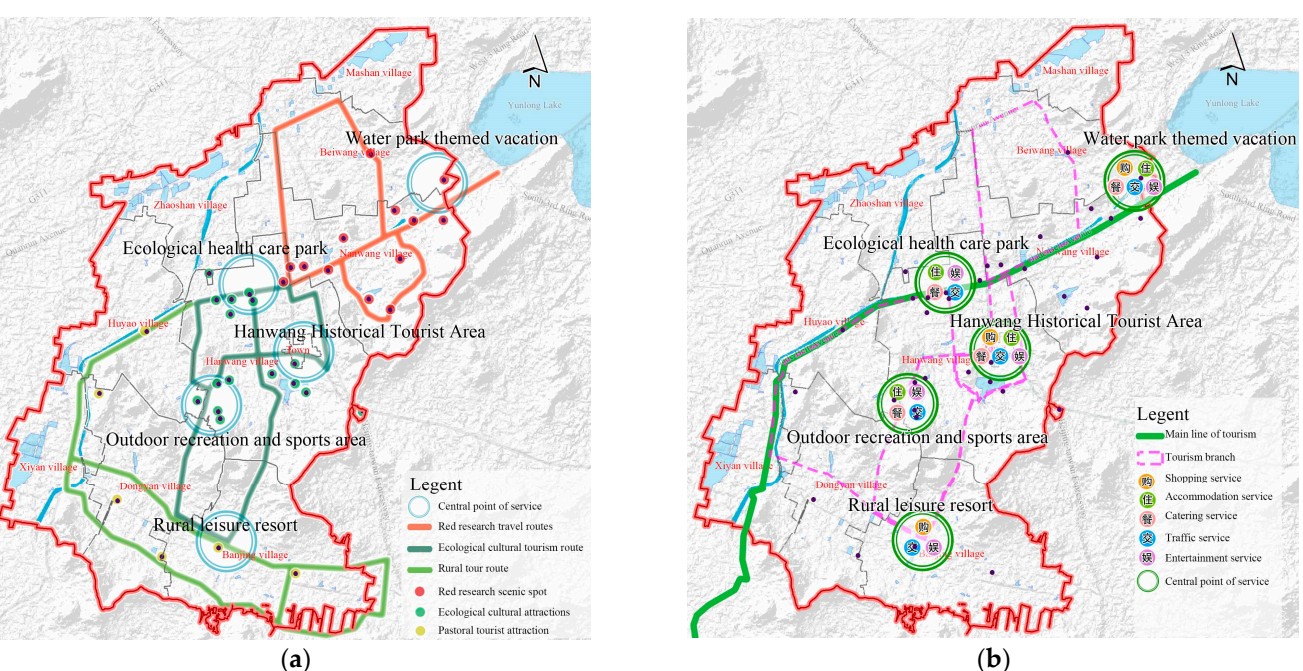

**Figure 10.** Tour routes and service facilities. (**a**) Town area tourism route planning; (**b**) layout plan of tourism service facilities.

### 4.3.2. Improving Tourism Service Facilities

Road traffic service facilities need to build parking lot on the basis of ensuring the optimization of traffic lines. Parking lot ought to be arranged reasonably in the entrance, accommodation, catering and other service facilities concentrated area. The area scale of the parking lot should be calculated according to the number of tourists, average ride rate, vehicle capacity, vehicle area, the utilization rate and other relevant data. At present,

the accommodation facilities in Hanwang town only Xuzhou Paradise Hotel is arranged while the others are scattered. It is suggested to arrange accommodation facilities near the Pana Valley ski resort to meet the needs of tourists. In the south of the core scenic area of Hanwang Reservoir, residential hostel can be built to develop recreational fisheries that integrating fishing, catering, leisure and other entertainment projects. The special catering of Hanwang town should combine the characteristics of Han culture with the flavor of Xuzhou, which has a deep local brand and rich cultural connotation. Hanwang town can set up special food commercial streets in important tourist spots, such as Hanwang Scenic Spot and Xuzhou Paradise, to explore traditional dishes that blend culture. Shopping is one of the six elements of tourism activities, which can bring huge economic benefits. The each shopping area should be combined with the tourist route, which ought to be arranged at the last position of the travel route. Because tourists will have the impulse to buy souvenirs after enjoying the natural scenery and cultural heritage. (Figure 10b).

## 5. Conclusions

This study constructs the GEP accounting index system of Hanwang town and calculates the results of GEP. According to the characteristics of natural resources and regional economic development of Hanwang town, the value accounting index system is divided into three categories: product provision value, regulation service value and cultural tourism value, which are specifically divided into 14 indicators. This study calculated its total value at 1.165 billion CNY in 2020. The results of GEP make a good foundation for the use of natural capital and value transformation of natural resources.

Hanwang town has a huge ecological environment value whose value structure is not the same to the general town. Because its cultural tourism value is 820 million CNY, much larger than the mediation service value of 239 million CNY, and the product value of 106 million CNY. This paper establishes four path of ecological value transformation of Hanwang town. Additionally, through this path, GEP can be transformed into GDP that helps urban and rural develop ecology industry. This part of the research enriches the connotation and form of ecological industry development. The specific strategy includes the development of agriculture, forestry, animal husbandry, fishery and aquatic industries, the enhancement of ecological restoration value, carbon emission rights and pollution emission rights trading, and tourism development, which elaborate the ecological industry development of the Town. This study believes that the strategy of guiding ecological industry development through GEP accounting is an effective way to realize sustainable development.

**Author Contributions:** S.T. is responsible for the overall structure and framework of this paper, data calculation, formal analysis and writing—original draft preparation. J.G. is responsible for the industrial planning according to the GEP results and writing—original draft preparation. F.W. is responsible for data curation. X.J. is responsible for guiding the article structure and methodology adjustment. All authors have read and agreed to the published version of the manuscript.

**Funding:** This research was supported by the National key Research and Development Program "Research on the Development Mode and Technology Path of Village Construction" (No: 2018YFD1100200) and the Major research fund project in Jiangsu Collaborative Innovation Center for Building Energy Saving and Construction Technology "Research on green infrastructure construction and spatio-temporal evolution characteristics at the town scale" (No: SJXTZD21055).

**Data Availability Statement:** The original data of the research belong to the data of the third China Land Survey, which is confidential and cannot be disclosed. However, process data can be disclosed.

**Acknowledgments:** We want to thank all reviewers for their valuable advices on this study, which made the description of the research results more clear and reasonable.

**Conflicts of Interest:** The authors declare no conflict of interest.

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
