# Peer review of "Research on Township Industry Development under GEP Accounting—A Case Study of Hanwang Town in Xuzhou City"

_land, doi:10.3390/land12071455_

Round 1

Reviewer 1 Report

This research is likely to discuss the economic aspects of urban development. 

Ecological benefit delivered to human beings and built environment, including urban or rural environment, has been much discussed by scholars in urban planning, landscape field.  

In general, In this research, how do you define ecological benefit? It does not just only count the value separately like the way it did. It is a complexible mechanism. 

I understand that the authors were attempting to clarify and connect the ecological value with urban benefit. However, which index do you believe would be a benefit for transforming the value of ecological space in towns? And how?

It should be discussed by ecological landscape knowledge by LULC and other landscape metrics.

In specific, how do you detect/identify the ecological environment value accounting index and functional correlation? And how do you grasp the data for this point?

In lines 195-196, double-check with first-level indicators (table 2). It does not match each other. 

On table 2, on Agriculture Products, which formula that you use for value calculation? Double-check the Citation for each of them.

For cultural tourism, In line 201,  clarify the relationship between cultural tourism and natural landscape. By the way, it will light up your value quality index for cultural tourism, shown at the end of Table 2. 

Another concern is how to build up a Gross Ecosystem Product accounting index system involving the spatial distribution of ecological value.  Just by sketching it up by boundary/ categories? 

Author Response

1.Ecological benefit delivered to human beings and built environment, including urban or rural environment, has been much discussed by scholars in urban planning, landscape field.  

Yes!

2.In general, In this research, how do you define ecological benefit? It does not just only count the value separately like the way it did. It is a complexible mechanism. 

Ecological benefit refers to the law of ecological balance in production. It is the beneficial influence and effect of natural biological systems on human production, living conditions and environmental conditions. It is indeed a complex mechanism, which is related to the fundamental and long-term interests of human survival and development. However, GEP accounting is a maturing method in China, and in 2021 it has been included in the United Nations' latest system of environmental and economic accounting, the Framework for Ecosystem Accounting (SEEA-EA). It mainly has three accounting contents, one is to calculate the value of product supply, the second is to calculate the value of product adjustment, and the third is to calculate the value of cultural tourism.

3.I understand that the authors were attempting to clarify and connect the ecological value with urban benefit. However, which index do you believe would be a benefit for transforming the value of ecological space in towns? And how?

Through the analysis, it is found that the value of farmland food is low in the supply value, mainly because the price of ordinary food is regulated by the government. Therefore, high value-added food can be grown, such as organic rice, to promote value.There are many forests in the town, but there is a lack of understory planting industry, which needs to be explored.Hanwang Town has a number of coal mining collapsed abandoned ponds, which can be developed into fish farming areas.At present, the two springs in the town only have the role of ecological replenishment, they can be developed into commercial mineral water, and the ecological value is huge.

The regulation value is mainly related to ecological type and vegetation cover. The existing bare land of coal and stone mining in the town can improve the ecological benefits through ecological restoration. The value of carbon fixation, oxygen release, air purification and water purification can be sold through the established Emissions trading platform to promote the transformation of ecological value.

The ecological aesthetic value is mainly manifested through the income of tourism. To further improve the value of the indicator requires the expansion of new scenic spots, such as the combination of abandoned coal mining areas, mine management, and rotten villa area management. In addition, service facilities such as parking, accommodation and shopping should be increased to promote consumption and increase tourism income.

First level indicator

Second level indicator

Concrete measures(How?)

Four modes for value transformation 

Easy or not  

Product provision

Agricultural products

High value-added food can be grown, such as organic rice.

Ecological industry management

Forest products

Understory industry needs to be explored.

Animal husbandry products

Increase the special breeding industry.

Fishery product

Coal mining collapsed abandoned ponds can be developed into fish farming areas.

Water resources

The two springs can be developed into commercial mineral water.

Ecological energy

The existing bare land of coal and stone mining in the town can be fixed by ecological restoration. By increasing the extent of vegetation coverage, the value will be improved.

Ecological value promotion

X

Regulation services

Water conservation

Soil conservation

Flood control and storage

Carbon fixation and oxygen release

Building a carbon emission trading platform

Ecological resource index trading

Air purification

Building a emissions trading platform

Water purification

Climate regulation

By changing the ecological type of the surface through afforestation

Ecological value promotion

X

Cultural tourism

Tourism consumption

It requires the expansion of new scenic spots. In addition, service facilities should be increased to promote consumption and increase tourism income.

Ecotourism

4.It should be discussed by ecological landscape knowledge by LULC and other landscape metrics.

I agree with your suggestion. The GEP accounting in this paper is based on LULC(China's Third Land Survey data), and I have added some targeted discussion content.

5.In specific, how do you detect/identify the ecological environment value accounting index and functional correlation? And how do you grasp the data for this point?

There are three main accounting principles of GEP. First, the use value of ecological products is accounted for, including direct use value and indirect use value. Usually, non-use value such as the existence value of ecological products is not accounted for. The second is to calculate the value of final ecological products, excluding the value of intermediate ecological products; The third is to calculate the value of ecological products on the basis of the functional amount.The correlation of environmental functions and indicators is collated through analysis of other studies about GEP accounting.

6.In lines 195-196, double-check with first-level indicators (table 2). It does not match each other. 

Table has been corrected.

7.On table 2, on Agriculture Products, which formula that you use for value calculation? Double-check the Citation for each of them.

The value of agricultural products is the sum of all relevant agricultural products, and the data is from Tongshan Yearbook 2020, and Three-year Data Analysis Report of Hanwang Town, Xuzhou City.

8.For cultural tourism, In line 201,  clarify the relationship between cultural tourism and natural landscape. By the way, it will light up your value quality index for cultural tourism, shown at the end of Table 2. 

Thanks for your precious advice, I have clarified the relationship between cultural tourism and natural landscape in line 201.

9.Another concern is how to build up a Gross Ecosystem Product accounting index system involving the spatial distribution of ecological value.  Just by sketching it up by boundary/ categories? 

This is a very good question that used to bother me. My consideration is to find low areas through value distribution, and then find appropriate ways to increase the unit price, including changing the land use and changing the type of agriculture. I think the specific method should be combined with indicators to expand the narrative.

Reviewer 2 Report

Tong’s et al paper wants to explore the layout of industries in terms of econoimic values of services provided by each land type at a fine scale. It seems that the work was based on programs of territorial spatial planning. Surely, in territorial spatial planning, there are three spaces, living space, producing space, and ecological space. Waste from these living and producing sapces should be treated in order to protect environments, and ecological space should be protected. Often, ecological space includes grassland, forest, wetland, and water. Values of these land types are very diffent from one place to another and market value method can not be enough to assess them (especiall for no-use value such as existence value, bequest value, option value and altruism value and no-use value often being greatly larger than use value or non-marker value being greatly larger than market value). Overall, I do not think that your wrok can attack international readers and can contribute coordinated development between environment and economy, because you just listed a fact (three spaces) and concerned the existed environmental or economic problems. I suggest you should submit to a Chinese journal. If so, I think that you work may have some value for local decisions.

English can be improved.

Author Response

Please see the attachment for the authors' reply. 

Reviewer 3 Report

This paper used multiple statistical methods to account the gross ecosystem product in Hanwang Town in Xuzhou City, and the strategies for development and layout of ecological industry in the study area was proposed. However, some of the major methodology and results need further explanation, which are as follows. A major revision is recommended.

1. An intensive English proof read must be considered. The language editing is recommended.

2. Abstract: Line 20, the authors declared they built an accounting system for the GEP “applicable to village and town level”, after reading the manuscript, I did not find any particularity of this indicator system compared with the large-scale study area, please make a further explanation.

3. Figures: most of the figures in the manuscript are blurred and the text is illegible, the authors should adjust the font size and the resolution of figures. Figure 2, please reduce the number of hierarchical legends. Figure 2~Figure 7, remove the text in the top right corner of the figures. Figure 7~Figure 10, the text in the figure overlaps with each other, or overlaps the geographic boundaries or other elements, making it difficult to read and needs to be revised.

4. Introduction: the overall structure is not quite reasonable in this section, and major revisions are needed here. The second paragraph is too long to get the key points, and the content of background and existing researches needs to be simplified. In addition, remove the names of Chinese scholars in the text. Furthermore, the authors need to declare the reasons for choosing this study area.

5. 2.3.1 Selection of accounting index: As the authors mentioned in Line 99 or reference 24, the GEP could be counted in four parts. Why did the support value (service) not consider in the study, which is also an important component in GEP.

6. Section 4. Industry Strategy Discussion: the content of the discussion and the results cannot be well connected. How can the authors infer that these strategies should be implemented through the analysis in the results?

7. References: The format of references is not uniform, such as the format of the author’s name, the capitalization of words in the title, whether the journal’s name is abbreviated, and whether there is a DOI. In addition, too many Chinese literatures were cited, please supplement or replace them with the one in English.

8. Table 1 and Figure 1: in the first row, “town” was not a type in land use, please use “construction land” or “settlement land”, the corresponding text in section 2.2 should also be revised.

9. Table 2: The position of the formulas is wrong and cannot correspond to the text description in the table.

10.Line 225: Is this line redundant?

11.Table 3 and Table 4: the first column, where dose the “town” referred to?

12. Line 285-286: where are Figure 4-15 and Figure 4-16?

13. Section 3.3: the text corresponding to the method and formula are recommended to move to 2. Materials and Methods.

14. Line 354-355: Why does this sentence appear here? What does it have to do with formula (1)?

15.Line 509: delete the “(tourist attractions)”.

An intensive English proof read must be considered. The language editing is recommended.

Author Response

This paper used multiple statistical methods to account the gross ecosystem product in Hanwang Town in Xuzhou City, and the strategies for development and layout of ecological industry in the study area was proposed. However, some of the major methodology and results need further explanation, which are as follows. A major revision is recommended.

  1. An intensive English proof read must be considered. The language editing is recommended.

Yes, the English manuscript has been optimized and corrected carefully. 

2.Abstract: Line 20, the authors declared they built an accounting system for the GEP “applicable to village and town level”, after reading the manuscript, I did not find any particularity of this indicator system compared with the large-scale study area, please make a further explanation.

This expression has been modified. The author believes that the key to the study of GEP accounting in towns lies in more detailed land use data.Generally speaking, the accuracy of land use data interpreted by satellite remote sensing is not clear enough, while the land survey data is relatively clear. It is more appropriate to use survey data for village and town level studies.

  1. Figures: most of the figures in the manuscript are blurred and the text is illegible, the authors should adjust the font size and the resolution of figures. Figure 2, please reduce the number of hierarchical legends. Figure 2~Figure 7, remove the text in the top right corner of the figures. Figure 7~Figure 10, the text in the figure overlaps with each other, or overlaps the geographic boundaries or other elements, making it difficult to read and needs to be revised.

In order to make it easier to understand, the picture has been modified according to the suggestions.

  1. Introduction: the overall structure is not quite reasonable in this section, and major revisions are needed here. The second paragraph is too long to get the key points, and the content of background and existing researches needs to be simplified. In addition, remove the names of Chinese scholars in the text. Furthermore, the authors need to declare the reasons for choosing this study area.

According to the recommendations, the research background and current research content have been simplified, and the names of relevant researchers have been removed. Hanwang Town was chosen as the research area because it has a relatively good ecological foundation and is close to the main urban area. It has the potential to realize the transformation of ecological value.It also has the typical significance to study common ecological towns.

  1. 3.1 Selection of accounting index: As the authors mentioned in Line 99 or reference 24, the GEP could be counted in four parts. Why did the support value (service) not consider in the study, which is also an important component in GEP.

GEP accounting is a relative maturing method in China now. It mainly has three accounting contents, one is to calculate the value of product supply, the second is to calculate the value of product adjustment, and the third is to calculate the value of cultural tourism.The concept of GEP was put forward in 2013, and the reference 24 was published earlier, therefore including the value of ecological support services. However, later GEP accounting researches incorporated eco-support services into the regulatory services content, so it went from four parts to three parts.

  1. Section 4. Industry Strategy Discussion: the content of the discussion and the results cannot be well connected. How can the authors infer that these strategies should be implemented through the analysis in the results?

The industrial strategy part is really not easy to relate to the results of the analysis. Therefore, we changed the way of narration in Section 4 and tried to combine the research results with the industrial strategy. In order to facilitate understanding, we use indicators to discuss one-to-one correspondence.

First level indicator

Second level indicator

Concrete measures(How?)

Four modes for value transformation 

Easy or not  

Product provision

Agricultural products

High value-added food can be grown, such as organic rice.

Ecological industry management

Forest products

Understory industry needs to be explored.

Animal husbandry products

Increase the special breeding industry.

Fishery product

Coal mining collapsed abandoned ponds can be developed into fish farming areas.

Water resources

The two springs can be developed into commercial mineral water.

Ecological energy

The existing bare land of coal and stone mining in the town can be fixed by ecological restoration. By increasing the extent of vegetation coverage, the value will be improved.

Ecological value promotion

X

Regulation services

Water conservation

Soil conservation

Flood control and storage

Carbon fixation and oxygen release

Building a carbon emission trading platform

Ecological resource index trading

Air purification

Building a emissions trading platform

Water purification

Climate regulation

By changing the ecological type of the surface through afforestation

Ecological value promotion

X

Cultural tourism

Tourism consumption

It requires the expansion of new scenic spots. In addition, service facilities should be increased to promote consumption and increase tourism income.

Ecotourism

  1. References: The format of references is not uniform, such as the format of the author’s name, the capitalization of words in the title, whether the journal’s name is abbreviated, and whether there is a DOI. In addition, too many Chinese literatures were cited, please supplement or replace them with the one in English.

The references have been revised and, where possible, the relevant English literature has been replaced.

  1. Table 1 and Figure 1: in the first row, “town” was not a type in land use, please use “construction land” or “settlement land”, the corresponding text in section 2.2 should also be revised.

The expression has been modified to construction land.

  1. Table 2: The position of the formulas is wrong and cannot correspond to the text description in the table.

As suggested, the formula position has been adjusted.

10.Line 225: Is this line redundant?

No extra lines found.Possible editing errors.

11.Table 3 and Table 4: the first column, where dose the “town” referred to?

Town refers to the town center construction land, shown on the map.

  1. Line 285-286: where are Figure 4-15 and Figure 4-16?

It was a mistake and has been corrected. They should be Figure 6-(a).

  1. Section 3.3: the text corresponding to the method and formula are recommended to move to 2. Materials and Methods.

The formula is adjusted.

  1. Line 354-355: Why does this sentence appear here? What does it have to do with formula (1)?

This sentence should not be here, the content has been adjusted.

15.Line 509: delete the “(tourist attractions)”.

Have been deleted as suggested.

Round 2

Reviewer 1 Report

Dear Authors,

The research approach is good, but it is insufficient for decision-making due to ambiguity in the evidence and indicators.

The 5th and 9th point answer remains unclear. 

Author Response

1.The research approach is good, but it is insufficient for decision-making due to ambiguity in the evidence and indicators.

I have tried to relate the relationship between GEP accounting indicators and decision-making by discussing land use types in section 4. In this way, the decision-making may be sufficient.

2.The 5th and 9th point answer remains unclear.

⑤In specific, how do you detect/identify the ecological environment value accounting index and functional correlation? And how do you grasp the data for this point?

The correlation of environmental functions and indicators is collated through analysis of other studies about GEP accounting.The strength of correlation between ecological environment functions is determined by ecological research results, and some scholars have sorted out this relationship before. They summarized the importance of ecological functions of different ecological types. If some functions have obvious effects, they will be included in the GEP accounting indicators.Taking agricultural products as an example, farmland is generally seeded for agriculture, so it has the value. Forests, grasslands, construction land and bare land do not grow agricultural products and therefore do not provide associated value. Wild seeding (e.g. rice, lotus root) is possible in wetlands and shrub land, but it is rare and of low value. Therefore, although there are relevant functions, they are not calculated.The Price-related data can be obtained from the Xuzhou Statistical Yearbook and geographical data used for the study can be found at https://www.gscloud.cn/ .

⑨Another concern is how to build up a Gross Ecosystem Product accounting index system involving the spatial distribution of ecological value. Just by sketching it up by boundary/ categories? 

Product provision value is the total value obtained by summing direct product value, and then the spatial distribution of ecological value is made by area inverse calculation. In the process of calculating the regulation value, the result is settled according to the ecological category, and then divided by the area to get the spatial distribution of value. The value of cultural tourism is the sum of the results obtained by CVM and TCM method, and then the unit price is derived by the importance degree and radiation range of scenic spots. Product provision, regulation value and cultural tourism value are all related to ecological types. Therefore, it is normal that the final results of spatial distribution of ecological values are closely related to the boundary/ categories.

Reviewer 2 Report

I have suggested what you should do next previously.

Author Response

1.I have suggested what you should do next previously.

I am very grateful for your valuable advises, and the content of the study has been revised follow your suggestion.

Reviewer 3 Report

After revision, the manuscript by Tong et.al still have several problems that worries me. In particular, some of the problems mentioned in last round have not been completely revised, please address the comments bellow carefully.

1.  The English still need to be improved in the manuscript, and I suggest the help from a native speaker.

2.  Introduction: the second paragraph is still too long to get the key points, and it is recommended to segment it.

3.  Introduction: it is necessary to make further supplementation of the shortcomings of existing relevant studies.

4.  Figures: Most of my comments on the figures were not well addressed. For example:

(1) Most of the figures are still hard to read, please adjust the size and the resolution.

(2) Figure 3~Figure 10, increase the font size of the text in the figures.

(3) Figure 3~Figure 7, remove the text in the top right corner of the figures.

(4) Figure 7~Figure 10, the text in the figure overlaps with each other, or overlaps the geographic boundaries or other elements, making it difficult to read and needs to be revised.

5.  The first row of Table 1, change “Town” into “construction land”.

6.  The “3.1.1 subsection” is still in the manuscript, but I did not find 3.1.2 or 3.1.3. I believe Line 185 is unnecessary.

The English still need to be improved and I suggest the help from a native speaker.

Author Response

After revision, the manuscript by Tong et.al still have several problems that worries me. In particular, some of the problems mentioned in last round have not been completely revised, please address the comments bellow carefully.

  1. The English still need to be improved in the manuscript, and I suggest the help from a native speaker.

Yes, I made further language revisions.

  1. Introduction: the second paragraph is still too long to get the key points, and it is recommended to segment it.

I appreciate your suggestions and have segmented the long content.

  1. Introduction: it is necessary to make further supplementation of the shortcomings of existing relevant studies.

I have further supplemented the review of relevant studies about township industries under GEP accounting.

4.Figures: Most of my comments on the figures were not well addressed. For example:

(1) Most of the figures are still hard to read, please adjust the size and the resolution.

The images and resolutions have been adjusted according to the recommendations.

  • Figure 3~Figure 10, increase the font size of the text in the figures.

The font size has been increased.

  • Figure 3~Figure 7, remove the text in the top right corner of the figures.

The text has been removed.

  • Figure 7~Figure 10, the text in the figure overlaps with each other, or overlaps the geographic boundaries or other elements, making it difficult to read and needs to be revised.

Some useless road lines or boundaries have been removed to make the text easier to read.

5.The first row of Table 1, change “Town” into “construction land”.

Yes, I have changed “Town” into “construction land”.

6.The “3.1.1 subsection” is still in the manuscript, but I did not find 3.1.2 or 3.1.3. I believe Line 185 is unnecessary.

I am sorry for forgetting to delete the useless line “3.1.1 subsection”. So, there is no 3.1.2 or 3.1.3. Unnecessary line 185 have been deleted. 

Round 3

Reviewer 2 Report

No matter what you want to say, you just measured some services of ecosystems and want to transform these ecological values into economic values. I admit your results have implications, but only for local study area.  

Author Response

1.No matter what you want to say, you just measured some services of ecosystems and want to transform these ecological values into economic values. I admit your results have implications, but only for local study area.  

Yes, in the fact,the purpose of this research is to calculate ecological value with GEP accounting method, and then use this value spatial distribution to guide the development of ecological industry,  promoting the transformation of ecological values. Personally, although the area of the study is a local town, the logic of GEP accounting and ecological values transformation strategy are also significant exploration for other studies.